# Youth Perspectives on What Makes a Sports Club a Health-Promoting Setting—Viewed through a Salutogenic Settings-Based Lens

**DOI:** 10.3390/ijerph18147704

**Published:** 2021-07-20

**Authors:** Susanna Geidne, Mikael Quennerstedt

**Affiliations:** Faculty of Medicine and Health, School of Health Sciences, Örebro University, SE-701 82 Örebro, Sweden; mikael.quennerstedt@oru.se

**Keywords:** organized sports, participation, HPSC, salutogenesis, drop-out

## Abstract

Sports participation has the potential to contribute to young people’s health. A prerequisite for young people to benefit from sports is that they stay in sports. Studies that consider both personal and contextual factors are needed to unpack the broader health-promoting potential of youth sports. The purpose of the study is to contribute to knowledge about the health-promoting potential of young people’s participation in organized sports by exploring youth perspectives on what makes a sports club health-promoting with a focus on health resources that young people consider important for sports club participation. For this cross-sectional study a brief survey was conducted with 15–16 year old students (*n* = 123) at two schools in Sweden, asking three open-ended questions about their participation in sports. The study used a salutogenic theory-driven analysis in combination with statistical analysis. Five health resources that young people consider important for sports club participation are revealed. On an individual, more ‘swimmer’-related level, these are personal well-being and social relations, including relationally meaningful activities, and on an organizational level, relating to the ‘river’, that sports clubs offer a supportive and well-functioning environment. For sports clubs to be health-promoting settings for young people and thus hopefully to reduce drop-out, we need a more sustainable approach emphasizing drop-in, *drop-through*, and drop-over as a continuous iterative process. We also need to consider the complexity of sports participation for young people, involving individual, organizational and environmental issues.

## 1. Introduction

The health benefits for young people of physical activity are irrefutable [1]. Participating in sports therefore has the potential to contribute to young people’s health by increasing their physical activity [2,3] and their mental and social health, as well as developing life skills related to well-being outcomes [4,5]. A prerequisite for benefiting from sports in this way, however, is that young people stay in sports [6]. Unfortunately, participation in club-organized sports often peaks at around 11–13 years of age before declining throughout adolescence [2,7,8].

These unambiguous benefits of sports participation can, however, also be questioned from a health-promotion perspective. Sports clubs certainly have the potential to be health-promoting, but not solely because young people are physically active when doing sports [9,10]. Instead, the entire setting of sports participation, from individual to organizational factors including issues like favorable conditions for recruitment and retention, can potentially be seen as health-promoting for young people. At the same time other aspects of sports participation like pressure, bullying or social injustice can be detrimental for young people’s health, even if they are sufficiently physically active.

In order to unpack the broader health-promoting potential of youth sports more studies are needed that move beyond single-discipline perspectives and consider both individual and organizational factors [11]. These potentials are often underutilized and in need of further development [12], and focusing on why young people participate and why they drop out from sports can give some insight into what factors they consider important for staying in sports.

One way to solve the problem of youth sports being looked at from narrow, single-discipline perspectives is to ask young people themselves what they think, using a health-promoting approach and in particular using a salutogenic settings-based perspective. Asking questions about what makes sports clubs health-promoting settings, including both what promotes and what hinders young people’s development towards health, can contribute to revealing young people’s perspectives on issues that link individual and organizational characteristics in the setting of the sports club as a whole. In this way, health resources that young people consider important for sports club participation can be identified contributing to the field regarding the complexity of the health promoting potential of sports participation for young people.

The purpose of this study is accordingly to contribute to knowledge about the health-promoting potential of young people’s participation in organized sports by exploring youth perspectives on what makes a sports club health-promoting with a focus on health resources that young people consider important for sports club participation. The study will also provide recommendations for sports clubs regarding drop-out, drop-over and drop-through.

### 1.1. Background

There is plenty of research on young people’s participation in sports, examining such things as who participates (or does not), why they join, how they benefit, why they remain and why they drop out. These studies use different disciplinary perspectives, such as psychology, focusing for example on individual motives (e.g., [13]); sociology, focusing on organizational or policy aspects (e.g., [14]); pedagogy, focusing on learning as both a motivation and an outcome (e.g., [15]); as well as physiology, focusing on physical health benefits (e.g., [2]). Missing, however, to a large extent are studies exploring young people’s own perspectives on what makes a sports club health-promoting. This section will present previous research on young people’s participation in organized sports, and then research on how young people want sports to be organized. The first part is on sport participation and particularly organized sport, but since there sometimes is no clear cut between different types of organization of sport, and that knowledge from different types of organization could give valuable insights the part could include a mix. The second part is focused on how the young people want sports to be organized, also here no clear cut between different types of organizing. Finally, in this section we present how a salutogenic settings-based approach helps us further explore the health-promoting potential of sports clubs.

#### 1.1.1. Young People’s Participation in Sports

In many European countries, about two-thirds of children and adolescents participate in organized sports clubs [8]. In Nordic countries the proportions are even higher [16,17]. There are group differences, however, with boys having higher participation than girls [8], and those who reported higher parental income being more likely to participate [17]. During adolescence, participation in organized sports decreases [2,7,8,18]. Studies of young people’s participation show that they often become involved in sports at an early age, often in the company of, or supported by their parents [19]. Parents are crucial for initiating sports participation—drop-in—and continue to be important, but in different ways throughout adolescence. Lack of parental support has however been identified as a barrier to participation in sports [20]. Rottensteiner and colleagues [21] show that parents are less important for withdrawal from participation during adolescence, even though negative parental pressure can be a reason for drop-out [22]. Continued participation is further shown to be closely related to (i) fun and enjoyment, (ii) social aspects, and (iii) issues of competition.

Fun and enjoyment are emphasized in many studies as crucial for young people’s sports participation (e.g., [19,22,23]), but as Skille and Østerås [23] discuss, what fun means can vary. Fun can be highlighted as an exclusive reason for participation [22] or be related to other factors, such as meaningful activities, learning and development [23], and perceived competence [22], or it can be inseparably tied to social dimensions and meaning [19,24]. A lack of fun and enjoyment is further highlighted as a major reason for drop-out [22]. Social aspects are also highlighted as important factors for participation in organized sports [5,19,24,25]. These include the importance of friends and peers and the possibility to create new social networks [26], or even to consider the club as a sort of family [27].

A debated question in research concerns the young people’s perception of competing in their sports. Some studies conclude that competing can be a challenging and fun aspect of sports for young people [24,28], boys as well as girls [23], and that learning and development could be shown to occur in matches [15,19]. In contrast, some studies show that in some cases a reason for drop-out can be that organized sports are too competitive [16,29].

Enjoyment, social relations and competition are emphasized broadly by different young people, but some factors also differ, depending on who the participants are and where they live. For example, age and gender are connected to several youth sports experiences [30]. One example is maintaining a slim body and getting fitter, which seems to be a more important factor for participation among girls, and during adolescence, rather than childhood [19,26,31]. Additionally, other contextual factors like living in different neighborhoods, disabilities, being from a family with lower socio-economic status and a being a recently arrived immigrant seem to influence participation [32,33,34,35].

#### 1.1.2. How Do Young People Want Sports to Be Organized?

Recurrent themes in the scientific literature regarding how young people want sports to be organized have to do with (i) flexibility and variety, (ii) a focus on social relations, (iii) the importance of other activities than the specific sports, and (iv) structural and physical environmental factors of organizing sports.

Several studies conclude that young people want a balance between structured and unstructured activities [27,36,37]. On the one hand, more flexible, less structured opportunities for physical activity are asked for by young people [36]. On the other hand, structure is needed for young people to have positive sports experiences [30]. Variety can also be created between different activities, such as by regularly modifying training sessions, to support creativity and give participants multiple opportunities to experience success [21]. Having different types of sports activities to choose from can, according to Skille and Østerås [23], make more young people enjoy sports. However, seeking variety and dropping out of sports can also be seen as a normal expression of young people’s changing interests and preferences during adolescence [38]. Although, dropping out of sport can mean starting another activity at the same time—drop over—or completely drop out [32,39].

Flexibility and variety can further be introduced to tailor organized sports to young people’s different requirements [24]. Here, competing can again be taken as an example, since the question might not be whether to compete, but rather how to compete. Säfvenbom and colleagues [28] conclude that competition is not a problem per se. Competition that strikes the right balance between challenges and success and is also related to doing one’s personal best can contribute to sense of achievement [19]. Andersson [15] even concludes that a typical match situation can have a pedagogical function if the expectations and seriousness of the match are adjusted to suit the participants.

A focus on social relations within the club is a crucial aspect of how young people want sports to be organized [19,24,26]. In this scenario, coaches are important actors, being the ones who directly deal with the young people in everyday practice. Cronin and Allen [4] argue that coaches have important responsibilities such as helping participants develop life skills, and providing a supportive environment and a safe, motivational climate for sports participation [21,25,40]. Rottensteiner and colleagues [21] highlight the importance of other activities than the specific sports, and state that it is important that coaches know their athletes outside of the club, and that they show interest in their lives in general. Stimulating participation in social activities and providing opportunities for volunteer work can, according to Deelen and colleagues [41], also be ways to keep young people in organized sports.

Finally, there are the more structural and physical environmental factors of organizing sports. Young people want the sports activities to be scheduled at convenient times, for example in relation to school hours, and also if possible, in the same facilities [36]. The sports activities can even be organized to accommodate other leisure-time activities, with flexible competition and training schedules [41], or having youth coach activities and participation in one’s own sports activities being scheduled together [42]. This flexibility in scheduling leisure time could reduce conflicts between sports clubs and school, for example, which otherwise, according to Temple and Crane [11], could lead to drop-out. Additionally, a collaboration between different sports clubs could encourage switching between different clubs and allow young people to change sports or clubs, rather than dropping out [29]. A collaboration between schools and sports clubs could also provide more affordable opportunities [20], and Humbert and colleagues [36] state that physical and economic factors, like having safe and clean facilities and reasonable costs for membership, need to be considered in organizing sports.

In conclusion, our review of research revealed that in many European countries a majority of young people participate in organized sports, and that participation is closely related to enjoyment, social aspects, and issues of competition. We further highlighted that variation, social relations, and environmental factors are essential to how young people want sports to be organized for them. It is a complex picture that is emerging and, as a consequence, single-perspective research can only provide a small piece of the puzzle. In response to this, many studies emphasize the use of multilevel perspectives in investigating and developing sports activities [22,38,43,44,45]. What is missing is thus an approach that takes more aspects into account and at the same time focuses on the relations between these different aspects of young people’s participation. We therefore suggest using a salutogenic perspective in combination with a health-promoting settings-based approach to sports clubs. A salutogenic settings-based approach can contribute with a broader perspective on health and the relations between different levels (from individual to policy level, including organizational) on how to work with sports club development.

#### 1.1.3. A Salutogenic Settings-Based Approach to Health-Promoting Sports Clubs

Settings-based health promotion originates from the Ottawa charter, where it is stated that health ‘is created and lived by people within the settings of their everyday life; where they learn, work, play and love’ [46]. This approach has been used in settings like schools, prisons, universities [47,48,49] and, from the early 2000s, also in sports clubs [50]. The theoretical basis for health-promoting sports club (HPSC) research has been thoroughly developed in recent years [51,52,53]. A HPSC can be described as a sports club that views all its activities through the lens of health promotion, including all club actions beyond promoting one single health behavior (like for ex. increasing physical activity or not using alcohol), involving participants in club decision-making, cooperating with external actors, and finally being aware that creating a healthy sports club is a continuous iterative process which must be based on the specific sports club’s needs [53].

In order to further unpack the health-promoting potential of young people’s participation in organized sports we turn to the salutogenic theory (e.g., [54,55]), as introduced by Aaron Antonovsky [56,57] and further developed for use in studies within sports and physical education [40,58,59,60]. A few studies have used a salutogenic perspective to study organized sports. For example, Thedin Jakobsson and colleagues [60] have studied who stays in sports clubs focusing on individuals, and Super and colleagues [40] have examined the roles of coaches in generating health-promotion potential.

A common theme in research adopting a salutogenic perspective is a resistance to defining health in a dualistic manner. Health is accordingly not to be viewed as the opposite of disease, i.e., something people either have or do not have. Rather, it is about different degrees of health, on a continuum, created and sustained in an ongoing process [55,61]. Wide-ranging individual, relational, social or cultural factors can thus potentially promote or prevent health development, and Antonovsky [57] uses ‘the river’ as a metaphor to discuss these characteristics of health. As Antonovsky [57] argued: ‘we are all, always, in the dangerous river of life. The twin question is: How dangerous is our river? How well can we swim?’ (p. 14). How we can understand health is consequently related to both our social, cultural and natural environment (the river), and our physical, social and mental resources as individuals (how well we can swim) [58,61].

Research must, according to Antonovsky, ask salutogenic questions, while taking both the ‘river’ and the ‘swimmer’ into account. Examples of salutogenic questions formulated by Antonovsky are ‘Why do people stay healthy?’ [62] (p. 35), or ‘What can be done in this community—factory, geographic community, age or gender group?’ [57] (p. 16). For us this involves asking questions about what makes sports clubs health-promoting settings, including both what promotes and what hinders young people’s development towards health, rather than exclusively focusing on what risks can be prevented through participation in organized sports. A salutogenic perspective thus helps us take into account individual and organizational aspects (and everything in between) of young people’s participation in organized sports.

In order to analytically unpack the health promoting potential we further use the salutogenic notion of health resources, which is described by McCuaig and Quennerstedt [58] as ‘different ways in which people from different backgrounds and in diverse contexts draw upon different resources to live a good life’ [58] (p. 119). A focus on health resources thus allows us to unpack the features of different identified resources by examining both young people’s (the swimmers) participation in organized sports and the characteristics of the sports clubs (the river) where young people participate.

#### 1.1.4. Purpose and Research Question

A way to solve the problem of youth sports being looked at from narrow, single-discipline perspectives is to ask young people themselves about what promotes or what hinders their development towards health in their participation. The purpose of the study is accordingly to contribute to knowledge about the health-promoting potential of young people’s participation in organized sports. This will be investigated by exploring youth perspectives on what makes a sports club health-promoting and focus on health resources that young people consider important for sports club participation. Our research question is accordingly:

What health resources characterize young people’s participation in organized sports and the sports clubs where they participate?

By combining a settings-based approach with a salutogenic perspective in a theory-driven analysis, we accordingly have the possibility to explore what health resources young people develop in sports club participation, thereby uncovering the health-promoting potential of sports clubs. These identified health resources will then also be quantified and used to discuss differences related to gender and participation in organized sports. The study will finally provide recommendations for sports clubs regarding drop-out, drop-over and drop-through.

## 2. Materials and Methods

### 2.1. Design and Sample

In 2019, Örebro University celebrated its 20th anniversary as a university. As a way of giving back to the local community, with which the university has had a good partnership during the years, secondary and upper-secondary schools were offered the possibility to ‘borrow’ a researcher who would give a talk about his or her research topic and what working in research is like. The schools could choose from different topics based on their interests and the available researchers. When three secondary schools chose to listen to a talk about health-promoting sports, we had an early opportunity to plan for this study. Two of the included schools had grade 9 (15–16 year old) pupils attend the event, and they became the sample for this cross-sectional study.

The schools were of similar character, being relatively large secondary schools located in municipalities up to 40 km outside of a larger city in central Sweden. At the first school, all grade nine pupils present that day participated (*n* = 73). At the second school, a few persons in various classes were absent to take part in another activity (*n* = 50).

The speaker (SG) presented issues like who participates in sports clubs (differences regarding age, gender, ethnicity, SES, etc.). This led to discussions on whether they recognized themselves in the patterns and if they thought it should be this way. The next topic concerned whether participation is good or bad for your health and why, for the most part using pictures showing different experiences such as joy, friends, and being outdoors, but also eating disorders, exclusion, and injuries. The speaker then focused on how to do research, and as a part of this, the pupils were asked if they wished to participate in a real research project. Included in this part was information on ethical questions related to their age, active consent, anonymity, and voluntary participation. The pupils were asked to answer the surveys individually where they were sitting. To fill out the paper surveys, they were given pencils which they could keep.

### 2.2. Survey

The survey had questions on gender as well as sports club participation, alongside three open-ended questions formulated as what characteristics of a sports clubs make them feel good or bad, and what a sports club can do to make them stay as long as possible.

The open-ended questions were inspired by McCuaig and Quennerstedt’s [58] advice not to ask directly about health, but instead to ask about living a good life. Furthermore, asking young people (or actually almost anyone) in Sweden about what makes a sports club health-promoting does not work well, since the most widely used Swedish term for health promotion often has a narrower and more everyday definition. Using open-ended questions gives respondents the opportunity to answer in their own words, instead of limiting them to predefined response options [16], which was considered appropriate because of the complexity of the concept of health promotion. Before the talks, the survey was piloted with young people in the same age group.

### 2.3. Analysis

A salutogenically guided theory-driven approach was employed in the analysis of the open survey questions [58]. To explore health resources in sports clubs as different ways that young people draw upon various resources through their participation in the clubs, and thus to identify what makes a sports club health-promoting, four analytical questions guided the reading of the transcripts. The questions (a–d), inspired by salutogenic theory in the sense that we focus on both young people’s participation in organized sports and the characteristics of the sports clubs where the young people participate, thus correspond to elements related to the swimmer and to the river, as well as to the relation between the two [57].

(a)What general components make young people feel good when participating in organized sports?(b)What components specific to the sports club make young people feel good when participating in organized sports?(c)What organizational components make young people feel good when participating in organized sports?(d)What components make young people feel good about how the sports club is organized for them?

The analytical questions were used in Step 1 of the data analysis, and both researchers read all answers individually several times. Using the analytical questions, both researchers formulated preliminary health resources in line with how a health resource can be delineated according to salutogenic theory [58]. The preliminary health resources became the basis for deliberation in order to make the resources into ‘something in common’ (see [63]). In a second step, the identified preliminary resources were accordingly discussed in depth in an iterative process where both authors were given the possibility to make judgements in relation to different alternatives and arguments. In this step the final agreement on the identified health resources was made. The five health-resource themes were then quantified by statement, and five dichotomous categories (one per health resource theme) were created and analyzed together with the descriptive data using bivariate statistics in form of non-parametric chi square test, conducted in IBM SPSS 27.

### 2.4. Methodological Considerations

To do brief surveys as a part of a popular lecture, “borrow a researcher”, can be discussed in terms of strengths and weaknesses. The popular lecture about health and sport focused on widening the perspective of the pupils concerning these concepts, not saying how it should be, but discuss what it might look like. A weakness could thus be that the pupils wrote what they thought we would like to hear. However, relating to the answers in the survey very few seem to have done that, since the examples from the presentation were not visible in the young people’s responses.

All pupils who were present the day of the popular lecture participated in the brief survey, irrespective of them being current, former, or non-participants of sports clubs. Considering that this study had around 90% of the pupils that had been members of a sports club, and around 50% still being members, the group were similar to Swedish young people in general.

The fact that boys’ answers were shorter and less comprehensive than the girls’ could, as emphasized by Persson et al. [16], be a problem when doing comparisons based on gender. This could result in boys’ perspectives on the issue being less clear than girls. Open-ended questions can in these circumstances be more suitable for girls than boys, and for a more comprehensive study, both fixed- and open-ended questions could be useful.

## 3. Results

### 3.1. Descriptive

The sample consisted of 123 participants aged 15–16 years (64% girls). At the time of the survey around half were members of one or more sports clubs (48% girls, 56% boys) and a further third had previously been a member of a sports club (41% girls, 34% boys). Only 11% of the participating girls and 10% of the boys had never participated in organized sports. Of the participants, 107 chose to write statements answering the open questions; these were a majority of the current participants (*n* = 59), and also included former participants (*n* = 41) and non-participants (*n* = 7).

### 3.2. Health Resources

In the analysis five different health resources were identified, covering the health-promoting potential of sports clubs in relation to the swimmer, the river and the relation between the swimmer and the river. The health resources can be seen as lying on a continuum from individual (the swimmer) to organizational (the river) (Figure 1). Below, the health resources are presented and illustrated with quotes from the participants. At the end of the section a quantitative overview is presented.

#### 3.2.1. Personal Well-Being

As a health resource more related to ‘the swimmer’, personal well-being is about enjoyment and feeling good about participating in organized sports.


*“It makes me feel good and I want to become a professional football player.*



*Fun to be a part of”*


Enjoyment is thus about having fun here and now, but also about a more long-term feeling of motivation and the pleasure of meeting other people. Further aspects of this health resource are doing activities you like, that the activities are fair and there is not too much pressure, and not taking things too seriously. The activities should also, according to the participants, be characterized by opportunities to develop as well as friendly competition together with teammates. However, there are also quite disparate notions of what the health resource of personal well-being can be, from winning, or rather not losing, to being physically active in order to feel healthy.


*“I feel bad when people say things like you’re no good at this.*



*If coaches say you’re good and get you to feel good.”*


#### 3.2.2. Social Relations

The health resource of social relations is characterized by good friends, teammates and coaches being friendly, but also by participants being encouraging and nice to each other. Sports clubs are described by some as a place to meet people, or specifically to meet new people, while others describe feeling good if they have a friend there. Sports clubs are sometimes referred to as almost like a family, and statements about experiencing a sense of community, team spirit, camaraderie, and friendship are common.


*“It’s very friendly, like a family.”*


Social relations are also described in terms of being cared about or told that you are an important teammate. The health-promoting potential of the sports club can further be threatened when people fool around or when other young people cannot handle losing a match without getting angry. Another problematic issue that was raised is not having anyone in the club to talk to if one has a problem, or feeling like one does not belong.

#### 3.2.3. Meaningful Activities

As a health resource for the participants, a sense of meaningfulness can be understood in terms of the relation between ‘the swimmer’ and ‘the river’. Here, meaningfulness is related to the activities and means that what they do when participating in the sports activity is worth the effort, which makes them feel good as a consequence. One illustration of what one young person wanted to do was:


*“Run, save, shoot, assist”*


Meaningful activities are also those where participants learn new things and develop within the particular activity, and where the coaches know what they are doing and help the participants to develop. Performing well in the sport is also considered a meaningful activity, while activities that are not organized or taught well are not considered meaningful.


*“Make sure you improve*



*Just send you out without any instructions”*


Meaningful activities also involve a sense of enjoyment, not only for oneself but for as many people as possible. They are meaningful if at the same time they also support social relations.


*“Doing fun things with the whole club*



*Other activities to generate camaraderie”*


Attending competitions and camps together, as well as travelling, sometimes even abroad, are other aspects of meaningful activities.

Variation can also be an aspect of meaningful activities as a health resource. Doing the same thing week after week is not what the young people in this study are looking for. A meaningful activity can be varied within participation in one sport, with regard to both the specific exercises to do during practice, and also how the practice sessions are structured overall.


*“Not always the same*



*Many different activities”*


Variation is also to do external activities, for example ones that are done together with teammates but are not part of the core activity of the club. Additionally, the degree of seriousness should vary for activities to be meaningful, i.e., sometimes more serious and sometimes less so. Activities should also be varied in relation to age and competence to be meaningful. Activities that for some reason lead to injuries or actually are considered dangerous to participate in are not meaningful activities.


*“Adapt to everyone’s level and not just rush ahead”*


#### 3.2.4. A Welcoming and Supportive Everyday Experience

An important health resource for making young people feel that they are participating in a fun, social and meaningful activity is a welcoming and supportive atmosphere in the sports club. This is often deliberately fostered by the coaches who train the young people in regular practice sessions. For example, the young people feel supported when they are involved in decisions and encouraged to do their best. Fair treatment is another important aspect of a supportive environment, i.e., that all participants are given the same opportunities and treated the same way, and that no one is excluded or favored in any way.


*“You should have the same opportunities if you are equally good.*



*Not only letting the best players play matches (everyone is valued equally)*



*So that you’re visible but not the centre of attention”*


Feeling welcomed also involves being treated with respect and given the same opportunities, no matter who one is or where one comes from. Regarding this, there is also a risk of coaches expecting more of participants than they can provide.


*“Not feeling like you have to perform at your best all the time, you can also have bad days”*


This welcoming atmosphere is not only important in relation to coaches; the sports club as a whole needs to offer a welcoming everyday experience.


*“It’s welcoming and it should be fun and pleasant to hang out with the people in the sports club.”*


In order to be welcoming and supportive, a sports club should accordingly be a safe place free from too much pressure and demands to prioritize this particular sport above everything else. Additionally included in a welcoming atmosphere is that the sports clubs show acceptance and understanding that people can be ill or have injuries.

#### 3.2.5. A Well-Functioning Sports Club

The welcoming and supportive everyday experiences that were emphasized in the previous health resource theme are also affected by how the club functions as a whole. In order to be a health resource, the sports club itself needs to be well-organized, making this resource closely related to the ‘river’. Here, structure, creativity, and a willingness to change are of the essence, and are evidenced in a well-functioning board and active work to develop the club.


*“Desire and motivation to develop the club*



*Doing the best they can”*


A well-ordered club is central to this health resource. For example, when it comes to the scheduling of practice sessions, it is important that sessions are not often cancelled, and it should be made clear how many practice sessions one should reasonably be expected to participate in. Costs, especially when they are too high, are not conducive to a well-functioning club. Receiving a free snack break (fika in Swedish) when members practice or serve as referees can be a health resource.


*“When things are well organized”*


Another aspect is for people in the sports club to understand and accept that the club is a part of the young people’s everyday life, and communication with school and other leisure activities is crucial. A collaborative environment that clearly emphasizes that all clubs prioritize the best interests of the child is part of this resource. One example is having a support system to prevent ill health, and another is the sports club showing interest in its members.


*“Sports clubs should collaborate and communicate with each other*



*A club that makes sure that the sport in question doesn’t cause the members to feel bad, and if they do, takes care of these people”*


The physical environment is also part of a well-functioning sports club. Changing rooms and other spaces should not be messy.


*“Not disgusting and filthy...”*


It is emphasized by many young people in this study that the facilities should be fresh and clean, to characterize a sports club where they feel good and therefore could be a health resource for young people.

### 3.3. Gender Differences

Given the quite large amount of qualitative material, it can be possible to quantify the data and see if doing so can add some further knowledge to the topic at hand. Here the analyzed health resources are related to who said what in relation to gender (Figure 2).

Social relations is the health resource mentioned by the most participants (59%), followed by a welcoming and supportive environment (55%). Girls and boys differ with regard to how much they mention all health resources, except for meaningful activities, which both groups mention equally. Significantly more girls than boys mention all of the health resources, except for personal well-being. However, girls contributed more statements overall than boys, on average mentioning 2.7 (vs. 1.7 for boys) health resources (*p* < 0.001). Regarding the participation status of the respondents, the only significant difference was that current participants in organized sports mentioned social relations to a greater extent than former participants and those who never participated (71% vs. 54% vs. 22%, *p* < 0.05).

## 4. Discussion

In this final section we will first present the main findings and discuss them in relation to the literature, then we will discuss them in relation to our theoretical lens and then last as recommendations that can be concluded as potential consequences from these findings. Our results reveal five health resources that young people consider important for sports club participation. These health resources are identified on an individual, more ‘swimmer’-related level, with personal well-being and social relations, including relationally meaningful activities, but also a more organizational level relating to the ‘river’, i.e., that sports clubs offer a supportive and well-functioning environment. These five health resources are also interrelated in many ways, for example in terms of enjoyment, activities being varied and promoting development, and spending time together with nice friends and coaches in an inclusive well-organized club.

Some of the findings are well known from earlier research. That young people participate in organized sports because it is fun is almost an understatement, however as Skille and Østerås [23] discuss, what fun and enjoyment mean may vary. Additionally, Persson and colleagues [16] discuss that enjoyment in sports is more complex in line with what Skille and Østerås [23] describe. Our findings indicate that fun and enjoyment are related to doing meaningful activities with friends in a well-organized and challenging environment, and that it is not fun when people fool around or when there is too much pressure. Friends are also important, and the sports club is a place where you can meet new friends, as well as spend time with old ones, friends you consider to be almost like family [5,19,24,25,27].

In relation to competition, we also find diverse perspectives, as earlier research has found [16,29]. Many of the young people in our study view competing with friends and teammates as a health resource, but they want everyone to be included in these matches and for there to be an understanding that you cannot perform your best all the time. As Säfvenbom and colleagues [28] argue, competition is not a problem per se. There is a risk that the concepts of competitiveness and competition will be conflated. An overly competitive environment is not desired, but competitive games are often considered fun and enjoyable, and thus to be reasonable club activities. There is a further risk that young people in sports clubs who are more focused on competition may receive more training from more competent coaches. However, not receiving as much instruction just because you will never reach elite level (or even want to) is a demoralizing thought; young people (and others) probably want to develop, learn, improve or be challenged on the level where they are.

Previous research has also emphasized that activities outside of regular everyday practice are important factors for staying in sports clubs [21]. These activities are also mentioned as a health resource in our study. These could be such things as attending camps and travelling, but also other types of external activities like having someone to talk to about problems that are not necessarily related to sports.

An issue that was mentioned in earlier research but almost not at all in our study is the physical health benefits of doing sports [19,26,31]. Humbert and colleagues [36] for example emphasize that all sports should create an environment that helps participants, especially young women, to feel comfortable with their bodies, but that this should not be the sole motivation for continuing with the activities. Parents are further not mentioned at all as health resources in our study, which agrees with the finding of Humbert and colleagues’ study [36] of the same age group. One reason for this could be that for young people in this age group, parents have receded somewhat into the background of their interest. As long as parents provide proper support and things run smoothly, the young people do not even consider them especially important.

In contrast to other studies, however, our study indicates the importance of inclusion, respect and fairness in their participation in sports. The young people in our study were 15–16 years old, an age when about half of the cohort has already dropped out of organized sports [2,8]. It might seem reasonable to assume that it is the more competitive ones and those who ‘were chosen for the team’ that stay, but somewhat surprisingly, a majority of the young people considered a welcoming and supportive club atmosphere to be one of the most important health resources for sports participation. There was furthermore no difference between the current and the former participants in this regard.

Another issue has to do with how the young people in the study expressed the relationship between the well-functioning aspects of the sports club and its health-promoting potential. Here, the difference between genders can be highlighted. With the exception of personal well-being, girls highlighted all the health resources we identified to a higher degree than the boys. They also wrote more and longer statements in the survey, presenting a more complex picture of what does or does not make them feel good in a sports club. This could indicate that issues in the club setting beyond just sports activities are even more important for girls than for boys but could also be explained by boys’ answers being shorter and less comprehensive than the girls’ answers, as also found by Persson et al. [16].

### 4.1. How Can We Understand This Result in Terms of a Salutogenic Settings-Based Approach?

Many studies on sports participation for young people describe the reasons for participation in sports as complex, interrelated, and something that should be explored as a whole [11,16,22,32,38,43,64]. To do this, previous research has for example used multilevel perspectives on how to develop sports [45], children’s rights perspectives [65], or programme implementation in empowering coaching programmes [66].

Using a settings-based approach to sports clubs, as we have done, contributes with knowledge about the entire participatory and organizational setting of club-based sports, and not least the relations between levels. By combining this with the salutogenic perspective, we have the further possibility to explore what health resources young people develop in sports club participation, thereby uncovering the health-promoting potential of sports clubs. In line with Thedin Jakobsson and colleagues [60] and Super and colleagues [40], we contribute to the field and add a salutogenic settings-based approach that takes into account both the swimmer (the participant) and the river (the sports club), as well as the interaction between them.

The potential of looking at sports clubs in this way shines light on the fact that health resources promoting young people’s participation can originate in one part of the club, affect activities in another part, and have consequences for a third part, often for the participants themselves. For a sports club to become more health-promoting for young people accordingly depends on linking the origin of issues to activities in a broader perspective, in order to find solutions *before* the issues have adverse consequences such as drop-out. The issues mentioned by the young people in this study as health resources promoting participation in organized sports often comprise broad and complex themes, like a good atmosphere, preventing injuries, variety in activities, and flexibility. On the other hand, if a salutogenic, settings-based approach is applied to all of a sports club’s coordinated activities, it can affect the whole club, because the causes of problems often come from different parts of the club and can be solved in other parts. Using narrow perspectives in isolation (such as searching for answers within the bounds of a single academic discipline or solving practical issues with policies that just look at a single phenomenon such as injuries) can often cause one to miss the bigger picture. Like all settings, sports clubs are complex systems and must be treated as such [67].

Looking at young people’s participation in organized sports through a salutogenic settings-based lens can help us to discover how the perspectives relate to each other, rather than just be limited to the specific perspectives alone. Many previous studies (e.g., [21,41]), for example on personal reasons for drop-out, have provided recommendations on how sports clubs should be organized to keep young people active in sports. Even though the recommendations are well thought through, they originate from single-perspective research. We argue that using a salutogenic settings-based approach when providing recommendations for developing sports makes it possible to contribute to more comprehensive recommendations building on a multi-perspective approach.

### 4.2. Recommendations for Sports Clubs Regarding Drop-Out, Drop-Over and Drop-Through

As potential consequences of our findings in relation to our salutogenic settings-based lens our study suggests that in order for sports clubs to be health promoting for young people, and thus hopefully to be able to retain more young people for a longer time and minimize drop-out, they should take a comprehensive approach to the entire sports club setting. This involves looking at individual aspects (regarding the swimmer), organizational and environmental aspects (regarding the river), and relational aspects (regarding both) and working comprehensively with the activities, aims, and purposes of the club [10].

Drop-out, particularly during the teenage years, is a huge problem for sports clubs, organizations, and decision-makers. Even though dropping out of any particular activity is probably a natural occurrence for almost all participants, for instance because their life situations change, it is important to support every young person who wants to continue with sports. Perhaps using the concept of *drop-out* in relation to a specific club or a specific sport is not even the best way to view the issue. Fraser-Thomas and colleagues [32] and Humbert and colleagues [36] suggest that when young people want to end their participation in one sport or sports club there should be a comprehensive idea about how to help them to seamlessly move into another sport or activity, which they call *drop-over*. We further want to add, as our own suggestion, the possibility of an emphasis on *drop-through,* which deals with how sports clubs can offer different activities, meet different demands, or offer different ways to participate in the club within their own organization. This is a way to adapt the sports club to the young people rather than making the young people adapt to the club.

Our recommendation here is that since dropping out of a sport, according to Eliasson and Johansson [64], is a long-term process, there may be things that could be changed during this process to create more opportunities for young people to opt for drop-over or drop-through. Since having other things to do is the main reason for withdrawal [21], together with pressure and a lack of enjoyment [22,29], a first part of this process would be that sports clubs can offer different things to do in the club. This could take the form, for example, of providing an opportunity to scale up or down the number of practice sessions, or to change to a more or less competitive focus, as ways to move forward while remaining within the club, i.e., drop-through. Clubs could further offer other activities connected to the same environment, like volunteering or referee education, for example.

This could contribute to solving issues like the need to achieve flexibility and variety, as recognized by Schlesinger and colleagues [38] and Humbert and colleagues [36] as well as in our study. It could also be a way to lower thresholds for *drop-in* or drop-over by attracting different groups of young people who find other things enjoyable and meaningful than competing in sports. Offering a membership discount for young people who do volunteer work in the club can benefit both the young people with less economic support from home and others seeking variety or who are injured. Volunteering, according to Geidne [42], also benefits the organization itself by creating a sports club environment where members feel like they belong, are listened to, and find things arranged in a way that enables smooth participation. In addition, close collaboration between different sports clubs could make it easier to change between different clubs (drop-over) and actively help young people change sport or club rather than dropping out [29].

For a sports club to fulfil this potential, these recommendations also need to be built into the organizations, so that they do not depend on individual coaches, single projects, or a particular group of hard-working board members. To make clubs more sustainable, drop-in, drop-through, and drop-over need to be emphasized as a continuous iterative process, which is in line with the young people’s wish that the sports club should have a willingness to develop the club. An illustration of this concerns how to work with coaches in a club. The importance of coaches for sports clubs’ activities cannot be emphasized enough. Actions by coaches are included in all the different health resources, sometimes explicit and sometimes more implicit. Coach education is therefore always recommended, in order to develop coaches’ ability to create a motivating and exciting, as well as welcoming and supportive environment (cf. [21,40]). However, the answer is more complex than that, as the content of the education is obviously crucial, and equally important is the support that co-coaches and club management provide in striving together to achieve the aims of the club [66]. Coaches need continuous education and support and should never be left on their own. Sustainability depends on coaches being part of the shared vision, climate and structure of the club as a whole.

Of course there are limits to how much can be changed, and not all sports clubs can have activities that suit everyone. Therefore, even if drop-through is probably a better option, since it demonstrates that a club takes responsibility for its participants, drop-over is definitely better than drop-out, in a long-term perspective. Importantly, drop-over should not just be about a club first trying to keep its members by any and all means, and then, when at risk of drop-out, encouraging them try other sports [29]. Recommending drop-over can also be problematic for a sports club if there is no ongoing long-term collaboration with other sports clubs and different external actors. Collaboration between different sports clubs, and between sports clubs and schools, is also something mentioned by the young people as being a health resource. This is another of the areas where we should listen to young people’s views on what makes a sports club a health-promoting setting.

A final reflection on organized sports is that it differs in many ways, for example, how they work with recruiting and retaining, but taking a broader perspective on young people’s participation in sports, it seems like we need to further question what organized sports is, and what young people want it to be. Our findings are well in line with the conclusions from the literature (on different types of organized sports), that different forms [23], different actors working together [20] and also offering a variety of structured and unstructured opportunities [27,36,37] can lead to more young people participating in sport, and as we conclude make them feel good in the setting of organized sport.

## 5. Conclusions

For most young people, participation in organized sports is one piece of the puzzle of living a good life; the club is a place where you want to be listened to, cared about, and believed in, at the same time as doing movement with friends. Participating in sports is potentially so much more than just being physically active. The young people in our study seem to understand quite well that good (healthy) sports activities, which contribute to health resources relating to ‘the swimmer’ as well as ‘the river’, do not just happen. They recognize that how the activities are organized matters, but also that creating and sustaining that type of a sports club is a complex matter, and as long as they feel that the clubs are doing their best, they are satisfied with that.

If we are to solve the bigger issues of sports participation and drop-out, we need to adopt a more comprehensive approach in order to also understand drop-over and drop-through in sports. We also need to take into account the complexity of sports participation for young people, as it involves individual issues (relating to the swimmer) and organizational and environmental issues (related to the river). As we suggest, a salutogenic settings-based approach to sports clubs is one way to move forward and develop sports for sustainable futures.

## Figures and Tables

**Figure 1 ijerph-18-07704-f001:**
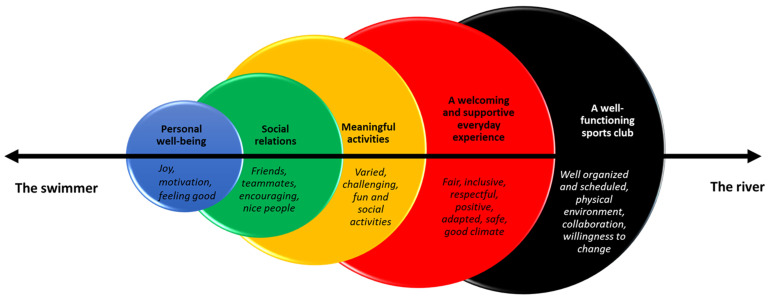
The five health resources on a continuum, illustrated using a salutogenic settings-based lens.

**Figure 2 ijerph-18-07704-f002:**
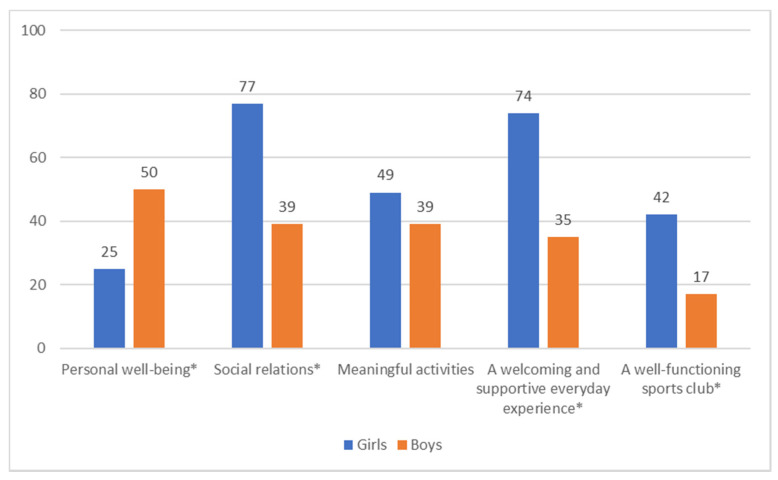
Proportion mentioning specific health resources divided by gender (in %, * *p* < 0.05, *n* = 244 statements).

## Data Availability

The data presented in this study are not publicly available but can be provided by the corresponding author in response to a reasonable request.

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
