# Peer review of "Youth Perspectives on What Makes a Sports Club a Health-Promoting Setting—Viewed through a Salutogenic Settings-Based Lens"

_ijerph, 2021, doi:10.3390/ijerph18147704_

Round 1
Reviewer 1 Report
GENERAL COMMENTS
While there are potentially interesting findings presented in the article, I still do not clearly grasp the purpose of the article and how this is related to ‘health’. The authors discuss sports clubs as potential health promotion settings for young people, but empirically the focus is not on the potential of sports clubs to be health promoting, rather on what constitutes favourable conditions for the recruitment and retention of young people in sports clubs – from the young peoples’ perspective. As such, it is taken for granted that sports clubs function as health promoting settings, which leaves me wondering how the ends really tie together and what is actually the purpose of the article: to examine favourable conditions for health promotion in sports clubs or to examine favourable conditions for recruitment and retention of young people. More clarity in this regard is needed throughout the article: e.g. when describing the purpose of the study, the literature review, the description of the results and in the discussion and conclusion. Also, the
SPECIFIC COMMENTS
INTRODUCTION
There is much interesting information in the introduction, but the structure could be improved to highlight even more the contribution of this article and to inform the reader more thoroughly about the relevance of the sub-paragraphs. Questions that should be tended to in this regard:
-How should the reader understand the sub-headline 1.1 ‘Background’? Background for what?
-The authors cite studies both on sports participation broadly and organised participation in sports clubs. This needs to be explained to the reader and the authors should make sure that it is explicit to the reader when studies are made on sports participation broadly and sports participation in clubs explicitly.
-What is meant by ‘organized’ (sub-headline 1.1.2)? The authors mention organized sport, but this paragraph is about a different perspective on ‘organization’ of sport.
More generally, the authors need to tie the pieces better together and help the reader more to understand the role and relevance of each sub-paragraph: what is the common thread throughout the introduction?
As it stands, there is also a number of repetitions in the introduction, e.g. it is repeated a number of times how the salutogenic settings-based perspective is chosen. The authors should read carefully the introduction to reduce such repetitions. Perhaps an improved structure can help accommodate this.
I am also a bit confused about the purpose of the study. On p. 2 it is stated that “The overall ambition of this study is accordingly to contribute to knowledge about the health-promoting potential of young people’s participation in organized sports by exploring youth perspectives on what makes a sports club health-promoting.” But then on p. 5 the research question is stated as: “What health resources characterize young people’s participation in organized sports and the sports clubs where they participate?”. It seems there are differing descriptions of what is actually the purpose of the article. Could the authors clarify?
Regarding the sentence: “Unfortunately, 31 participation in club-organized sports peaks at around 11–13 years of age before declining throughout adolescence [2,7].” (p. 1). Can the authors specify whether this claim is universal or country-specific. I would suspect that there are significant country differences.
The authors pose the question: “The question then is whether they start another activity at the same time – drop over – or completely drop out 126 [32].”. They do not provide an answer, but maybe the work of the Norwegian researcher Ørnulf Seippel can be helpful in this regard. Since the article originate from a Scandinavian context, the authors should be able to read the report in Norwegian that can be accessed through this link: https://samfunnsforskning.brage.unit.no/samfunnsforskning-xmlui/handle/11250/177519).
MATERIALS AND METHODS
In section 2.1., it is stated that the pupils answered the questionnaire after an introductory lecture in which topics were touched upon relating to the topics of the questionnaire. Could the authors reflect on whether this could have had an effect on the answers collected from the pupils?
In the same section the setting for the data collection was described, but it is not commented upon by the authors to which extent – and with which reservations – the results from this study could be perceived as general to young people. This would strengthen the paragraph, as it informs the reader about potentials and limitations in the data.
In section 2.2., it is stated that the pupils were asked questions about the characteristics of sports clubs that made them feel good or bad. Were pupils also asked this question, provided they were not members of the sports clubs? And are such answers relevant? Why?
In the same section, the pros of using open-ended questions are written out, but the potential limitations of this approach are not described. Could the authors address these as well?
In section 2.3., the questions asked by the researchers from a salutogenic perspective (a-d) are stated, but the wording of the questions to the pupils are not included. This would be very informative for the reader to be presented with these as well. E.g. it allows the reader to make own judgements about the relevance and usefulness of the questions asked.
RESULTS
Writing the results section presents a fine line between generalising from results and taking into account contradictory statements. I urge the authors to go through the description again to make sure that the phrases used accurately describe the statements given. E.g. I find it hard to believe that one can generally say that: “Variation is another aspect of meaningful activities as a health resource. Doing the same thing week after week is not what participants are looking for.” (p. 9). Clearly, I do not have the material at hand, so I might be wrong in context to your material, but from other studies it is clear that sports participants are divided on this: some are very keen to specialise in one or a few activities, while others prefer to try a broader range of activities. The same reservation is present from my side regarding a generalising statement such as: “Instead, the facilities should be fresh and clean, to be a health resource for young people” (p. 10). Thus, are there in your material contradictory statements, for instance in these regards? Please also check this more generally regarding other statements presented in the results section.
I understand the need for the authors to provide examples from the data through the use of direct quotes from the pupils’ responses. However, in the current state, the included quotes are not always directly linked to the description provided and they worsen the reading flow. Could they be moved to a table with exemplifications for the general descriptions? Or in some other way be placed so they are not ‘in the way’ when reading the text?
DISCUSSION
Regarding the statement, “Here the difference between genders can be highlighted. With the exception of personal well-being, girls highlighted all the health resources we identified to a higher degree than the boys. They also wrote more and longer statements in the survey, presenting a more complex picture of what does or does not make them feel good in a sports club. This could indicate that issues in the club setting beyond just sports activities are even more important for girls than for boys” (p. 12), the authors should consider also the methodological explanation that girls/women often are more inclined to and more through when answering questionnaires in general. This could also help explain this finding.
Section 4.2. presents discussions about drop-out, drop-over and drop-through which are certainly interesting topics. However, I feel this section is not sufficiently linked to the data applied in the article, which cannot inform us about these three concepts. Thus, the content seems often far-removed from the findings.
The discussion section lacks a methodological discussion to inform the reader about potentials and limitations in the data material.
CONCLUSION
Contributing to the confusion regarding the purpose of the article, the authors in a sentence: “The young people in our study seem to understand quite well that good (healthy) sports activities, which contribute to health resources relating to ‘the swimmer’ as well as ‘the river’, do not just happen” explain ‘good’ activities as ‘healthy’ activities. I am sceptic about this equating these concepts, and, as such, the conclusion adds to my confusion of the overall focus of the article and the use of the ‘health’ concept.
Author Response
Dear reviewers,
We would like to thank both reviewers for taking the time to review our manuscript and giving us valuable feedback. Below, we will answer to all of your comments and suggestions and tell what we have done to change the manuscript in relation to them. The changes are also made visible by track change function.
Susanna and Mikael

Reviewer 2 Report
This study explores the health-promoting potential of young people’s participation in organized sports. The topic of the paper is potentially interesting and attractive for professionals in the sport field and the healthcare field. However, I consider that the manuscript has important flaws and should be strongly improved.
First, I would like to kindly suggest to the authors previous work made by Downward about sports participation in Europe, the influence of parents, etc. [For example: Downward, P.; Lera-Lopez, F.; Rasciute, S. The correlates of sports participation in Europe. European journal of sport science, 2014, 14, 592-602]. In addition, there is useful information in the Special Eurobarometer on Sport and Physical Activity by European Commission [European Commission Special Eurobarometer 472 Report—Sport and Physical Activity. Available online: https://op.europa.eu/en/publication-detail/-/publication/9a69f642-fcf6-11e8-a96d-01aa75ed71a1].
Second, the methods could include more specific information. Was any software used for qualitative and quantitative data analysis? What was the procedure for qualitative data analysis? Was codification used? In this type of study in which an interpretation of reality is done, the characteristics of the researchers (expertise, experience, number, etc.) are fundamental. Please include a detailed description of these aspects. Thus, there are still many issues to explain for allowing replicability of the research. Perhaps, a new sub-heading about data analysis is also necessary here.
Third, I would like to suggest including some figures or tables after every section in the results, summarizing the main findings made by the authors. This would greatly help the potential readers to structure the results and understand them.
Fourth, in the conclusions, the authors should include four compulsory elements: (1) general summary of the article, its results, and findings, (2) implications and recommendations for practice, (3) research limitations, (4) suggestions for future research. Please add the missing information.
Specific comments
[In-text citations and references] Please be careful with the citations since there are numerous errors. Many in-text citations (the number of the citations) do not match with the references (the number of the reference).
[Figure 1] Is difficult to read. It is necessary to increase the size of the font.
[Page 11, lines 447-448] “Given the quite large amount of qualitative material, it can be possible to quantify the data”. Did the authors perform both qualitative and quantitative analysis? This should be clearly stated in the methods section. In addition, regarding Figure 2, was a difference of means conducted? Did the statistical differences appear between all groups? If yes, more information about it is required (statistical differences between all/some groups, specify them, tests performed, values, etc.).
The comments are expected to be useful.
Author Response
Dear reviewers,
We would like to thank both reviewers for taking the time to review our manuscript and giving us valuable feedback. Below, we will answer to all of your comments and suggestions and tell what we have done to change the manuscript in relation to them. The changes are also made visible by track change function.
Kind regards,
Susanna and Mikael

Round 2
Reviewer 1 Report
Though the authors have made significant improvements in the revision of the article, I still have three concerns remaining that I feel need to be addressed:
1.
In spite of the revisions made, I still maintain my critique, that it is not clearly stated what the article examines – or, rather, it is falsely stated that the purpose of the article is to understand the health-promoting potential of young people’s participation in sports clubs. The article does not factually examine this, rather it examines favourable conditions for recruitment and retention of young people in organised sport and the health-promoting potential is taken for granted. The approach of the authors is relevant, but different from the stated purpose, and this is to me a serious concern, but one that could be addressed, but has not been done so by sufficiently in the revision. As an example, it is still stated in the abstract: “The purpose of the study is to contribute to knowledge about the health-promoting potential of young people’s participation in organized sports.” Why not state what is examined: favourable conditions for recruitment and retention of young people in organised sport? This is still meaningful and immediately understandable to the reader – and it would not require the number of further explanations added in the introduction.
2.
I am also puzzled by this reply from the authors: “We agree with the reviewer that it is a not an easy distinction between these concepts, we will try to explain how we have thought. There can be a bit of a mix of these studies, but focus is on organized sport participation (in a broad sense), however there is not always a clear cut between them, and studies investigating both could be relevant for an understanding of both – and especially in the period of life when some young people change from one to the other, and also for specific groups, like low SES and girls, who seem to prefer the unorganized way.” Since the authors – that is my reading at least – focus on sports clubs, it should be possible to read again the cited studies and differentiate findings according to whether they were collected in a sports clubs setting or not. This would really qualify the literature review, since there are similarities, but also differences regarding how to recruit and retain young people according to the organisational context.
3.
I still believe section 4.2. is (too) far removed from the data collected in the article. This section could have been written without the empirical data that the authors have collected. So, even though the authors now specify that they in this paragraph present: “…recommendations that can be concluded as potential consequences from these findings”, I still believe that it is a serious flaw to add such reflections that are based on findings from the data. Thus, the authors would need to create a clearer link to the data in order to give recommendations building on their findings. At current, they are just recommendations from other literature.
Author Response
Dear editor and reviewer 1,
We would like to thank you for once again giving us valuable comments on our manuscript. We have answered to all three comments, one by one (copied in below).
Though the authors have made significant improvements in the revision of the article, I still have three concerns remaining that I feel need to be addressed:
1.
In spite of the revisions made, I still maintain my critique, that it is not clearly stated what the article examines – or, rather, it is falsely stated that the purpose of the article is to understand the health-promoting potential of young people’s participation in sports clubs. The article does not factually examine this, rather it examines favourable conditions for recruitment and retention of young people in organised sport and the health-promoting potential is taken for granted. The approach of the authors is relevant, but different from the stated purpose, and this is to me a serious concern, but one that could be addressed, but has not been done so by sufficiently in the revision. As an example, it is still stated in the abstract: “The purpose of the study is to contribute to knowledge about the health-promoting potential of young people’s participation in organized sports.” Why not state what is examined: favourable conditions for recruitment and retention of young people in organised sport? This is still meaningful and immediately understandable to the reader – and it would not require the number of further explanations added in the introduction.
Answer:
Thank you for letting us clarify this further.
We think this comment might have to do with, us and the reviewer coming from different science traditions in how to write purposes, we have tried to clarify the purpose in the abstract so that it will be clearer what our object of knowledge as well as object of study is. We have earlier just stated our object of knowledge in the abstract and thereafter added the object of study in the purpose section, now we have both in both places.
However, as the reviewer ask:
Why not state what is examined: favourable conditions for recruitment and retention of young people in organised sport?
We can´t state this, because this is not what the study have explored – but we agree with the reviewer that it could be a potential outcome of our study.
2.
I am also puzzled by this reply from the authors: “We agree with the reviewer that it is a not an easy distinction between these concepts, we will try to explain how we have thought. There can be a bit of a mix of these studies, but focus is on organized sport participation (in a broad sense), however there is not always a clear cut between them, and studies investigating both could be relevant for an understanding of both – and especially in the period of life when some young people change from one to the other, and also for specific groups, like low SES and girls, who seem to prefer the unorganized way.” Since the authors – that is my reading at least – focus on sports clubs, it should be possible to read again the cited studies and differentiate findings according to whether they were collected in a sports clubs setting or not. This would really qualify the literature review, since there are similarities, but also differences regarding how to recruit and retain young people according to the organisational context.
Answer:
Thank you for picking on us on this issue; it is a really important question and it made us think and put words on our reflections, which might have been implicit before. Going through the studies included in the literature review again, we can see that the majority of the included studies actually concerns participation in organized sports, often in the form of sports clubs, which is the most common way in for example Nordic countries, but not limited to sports clubs, because organized sports have different forms in different countries. Some of the studies (ex. Somerset and Hoare, 2018 and Skille and Østerås, 2011) more broadly examine participation in sports and Humbert et al., (2008) physical activity. But, when reading all of these, the findings concern young people who participate in organized sports (because they are included in the samples) and they are therefor valuable findings for this study. A few of the studies included in the literature review concerns sports and leisure organized in different ways than sports clubs, like Högman and Augustsson (2017) and Fredriksson et al., (2018). Together these different studies conclude that young people want organized sport to be something that is organized differently. We have added reflections on these issues of organized sports more explicit in the last part of the discussion, line 719-726.
3.
I still believe section 4.2. is (too) far removed from the data collected in the article. This section could have been written without the empirical data that the authors have collected. So, even though the authors now specify that they in this paragraph present: “…recommendations that can be concluded as potential consequences from these findings”, I still believe that it is a serious flaw to add such reflections that are based on findings from the data. Thus, the authors would need to create a clearer link to the data in order to give recommendations building on their findings. At current, they are just recommendations from other literature.
Answer:
Thank you for this comment. In our science tradition it is common that discussions are made, first closely related to the findings and earlier literature and therefor more, as we have written, as potential consequences of the findings. We have further added some links to our findings together with a repeated clarification that this is “recommendations that can be concluded as potential consequences from these findings”, but given the already quite long manuscript, we hope that the clarity of the findings sections could make readers see the links with our recommendations, which we as earlier mentioned are potential consequences of our findings, that is more a meta-discussion related to our findings and earlier literature, from using our theoretical lens.

Reviewer 2 Report
The authors have nicely addressed all the concerns raised by the reviewers in the previous round. They have properly justified and detailed the methods used. They have substantially improved the manuscript with important improvements in all the sections.
Congratulations on a nice job.
Author Response
Dear editor and reviewer 2,
Thank you.
Kind regards,
The authors